# Rainwater Harvesting Potentials in Commercial Buildings in Dhaka: Reliability and Economic Analysis

**Md. Rezaul Karim [1], B. M. Sadman Sakib [1], Sk. Sadman Sakib [1] and Monzur Alam Imteaz [2],***

[1] Department of Civil and Environmental Engineering, Islamic University of Technology (IUT), Board Bazar, Gazipur 1704, Bangladesh; rezaulmd@iut-dhaka.edu (M.R.K.); sadmansakib47@iut-dhaka.edu (B.M.S.S.); sadmansakib45@iut-dhaka.edu (S.S.S.)

[2] Department of Civil and Construction Engineering, Swinburne University of Technology, John Street, Hawthorn, VIC 3122, Australia

* Correspondence: mimteaz@swin.edu.au

**Abstract:** Despite numerous studies on residential rainwater tank, studies on commercial rainwater tank are scarce. Corporate authorities pay little heed on this sustainable feature. With the aim of encouraging corporate authorities, this study presents the feasibility and economic benefits of rainwater harvesting (RWH) in commercial buildings in the capital city of Bangladesh, where water authority struggles to maintain town water supply. The analysis was conducted using a daily water balance model under three climate scenarios (wet, dry and normal year) for five commercial buildings having catchment areas varying from 315 to 776 $m^2$ and the storage tank capacity varying from 100 to 600 $m^3$. It was found that for a water demand of 30 L per capita per day (lpcd), about 11% to 19% and 16% to 26.80% of the annual water demand can be supplemented by rainwater harvesting under the normal year and wet year climate conditions, respectively. The payback periods are found to be very short, only 2.25 to 3.75 years and benefit–cost (B/C) ratios are more than 1.0, even for building having the smallest catchment area (i.e., 315 $m^2$) and no significant overflow would occur during monsoon, which leads to both economic and environmental benefits. Though the findings cannot be translated to other cities as those are dependent on factors like water price, interest rate, rainfall amount and pattern, however other cities having significant rainfall amounts should conduct similar studies to expedite implementations of widescale rainwater harvesting.

**Keywords:** urban rainwater harvesting; non-potable water uses; mass balance model; payback period; Bangladesh

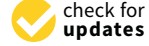

## 1. Introduction

In Dhaka, the capital city of Bangladesh, Dhaka Water Supply and Sewerage Authority (DWASA) is responsible for supplying potable water to the city dwellers. Among the total water abstractions, around 78% comes from groundwater [1]. The groundwater level in Dhaka is dropping by 2–3 m/year due to continuous over-extractions of water [2]. To reduce the dependency on groundwater in recent years, DWASA has undertaken several initiatives including rainwater harvesting, recycling of greywater, and others [1].

Rooftop rainwater harvesting in residential areas has received notable consideration as an alternative water supply source, and numerous research studies have been conducted all over the world [3–16]. In an urban context, two prime focuses of rainwater harvesting analysis are reliability and water saving potential [10]. All the above-mentioned studies reported decent achievements of reliability and water saving potential of rainwater harvesting (RWH) in urban areas.

Another important aspect of rainwater tank analysis is the economic feasibility, which is a key factor in decision-making by policy makers and building owners in adopting the RWH system. Rahman et al. [17] showed that payback for multi-story buildings in Sydney could be attained in the case of large numbers of users with a low discount rate.

Imteaz et al. [18] found the recovery of capital cost within 15 to 21 years under the increased water price for large rainwater tanks in commercial buildings in Melbourne. Zuo et al. [19] evaluated 267 RWH projects in Beijing and found 66.7% of the RWH systems economically beneficial. Matos et al. [20] investigated the economic feasibility of a rainwater tank in a commercial building in Broga, Portugal, and reported that with a 10% discount rate, depending on water uses and price, the payback period for the construction of the tank would vary from two to six years. Ghimire et al. [21] conducted a thorough life cycle assessment of a four-story commercial building with 1000 employees in Washington, D.C. (USA) and reported that, out of eleven life cycle impact assessment (LCIA) indicators, the rainwater harvesting system outperformed the traditional municipal water supply system for uses in toilets and urinals for ten of them. Nonetheless, the financial feasibility of an RWH system in urban areas depends on various factors such as initial capital cost of the tank, total water demand, water price, catchment area, tank size, rainfall intensity, as well as demand and uses of the harvested rainwater [22].

Very limited studies regarding rainwater harvesting in public or commercial buildings were found in the literature as compared to residential buildings, and no study has yet been conducted for the commercial buildings in Bangladesh. As mentioned by Ward et al. [23], major water demand in commercial buildings are for non-potable purposes such as cleaning and sanitation, for which a potable water quality standard is not required. The average yearly rainfall in Dhaka is about 2200 mm, 75% of which occurs during the monsoon period from June to October. With proper planning, this huge quantity of rainfall can be utilized as an alternative source of water, mainly for non-potable purposes for the commercial buildings in Dhaka. The unit price of water supplied by DWASA for commercial purposes is about 3.2 times higher than for domestic purposes [1]. Moreover, it has been made obligatory in the Bangladesh National Building Code (BNBC) [24] to preserve and harvest rainwater for every new building proposed on blocks having an area more than $300\,\mathrm{m}^2$. Due to a higher water price in the commercial sector and higher rainfall, RWH in commercial buildings is likely a promising alternative source for a non-potable water supply, resulting in significant monetary savings and also reducing the burden of the city's water supply. As such, it is important to investigate the feasibility, effectiveness, and potentials of water and monetary savings through a rainwater harvesting system in commercial buildings in Dhaka. The objectives of this paper were to explore the water saving potential and economic advantages of RWH in multi-story commercial buildings for non-potable water demands (i.e., car washing, toilet flushing, floor cleaning, etc.) under three climate conditions (i.e., wet, dry, and normal year) in Dhaka. The study findings are expected to contribute in filling the gap of novel information on the opportunity of placing an RWH system in commercial buildings and will also help policy makers in the water supply sector to understand why it is important to implement the legislative measures stated in the BNBC [24].

## 2. Materials and Methods

### 2.1. Study Area

Despite Bangladesh being a riverine country, its water demands for agricultural, residential, and commercial purposes are growing due to population and economic development [25]. The study area is Dhaka, which is the industrial hub and the capital city of Bangladesh, that with a population density of 44,000 persons/$\mathrm{km}^2$ [26] is the most densely populated city in the country. Presently, one-tenth of the population of Bangladesh and one-third of their urban population live in Dhaka [27]. The population of Dhaka is over 21 million, increasing at an annual rate of about 3.56% [28].

The average annual rainfall in Dhaka is about 2200 mm, 75% of which occurs during the monsoon (June–October) [16]. With proper implementation of rainwater harvesting facilities, rainwater as a source of clean usable water bears great potential in the city of Dhaka. Dhaka's average yearly precipitation during 1994–2003 tends to have ranged from 1500 to 2300 mm, that is, 1.50 to 2.30 $\mathrm{m}^3$ of rainwater per $\mathrm{m}^2$ per year collected from the Bangladesh Meteorological Department (BMD). A research conducted by the United

Nations Environment Program in 1982 found that with an average 72-inch rainfall and 1100-gallon reservoirs, sufficient water could be gathered in 12 h to support a household of six for 45 days.

According to the BNBC [24], water consumption for domestic purposes in residential buildings in Dhaka ranges from 40 to 260 L per capita per day (lpcd) and water consumption for commercial purposes (i.e., offices, industries, hotels, shops, hospitals, etc.) in Dhaka ranges from 3 to 450 lpcd. In most situations, Dhaka WASA's water supply system is dependent on groundwater. Around 78% of the water comes from groundwater sources and the remaining 22% comes from surface water [29]. Groundwater is abstracted using 887 deep tube wells and surface water is provided from the Shitalakshya and Buriganga rivers via 4 water treatment plants [29]. Dhaka WASA produced a daily water production capacity of 2550 million liters per day (MLD) over the period of 2017–2018 using 887 deep tube wells and 4 water treatment plants, including the Phases I and II Saidabad water treatment facility [29]. While this water scarcity has a direct impact on the city dwellers' water consumption, it is also taking a toll on the waste load on sewage lines. More importantly, economic extraction and consumption of the water supply are also a major concern for the suppliers and the consumers alike. Data regarding roof catchment areas and underground reservoir capacity were obtained for the analysis from five commercial buildings located in different parts of Dhaka (Figure 1).

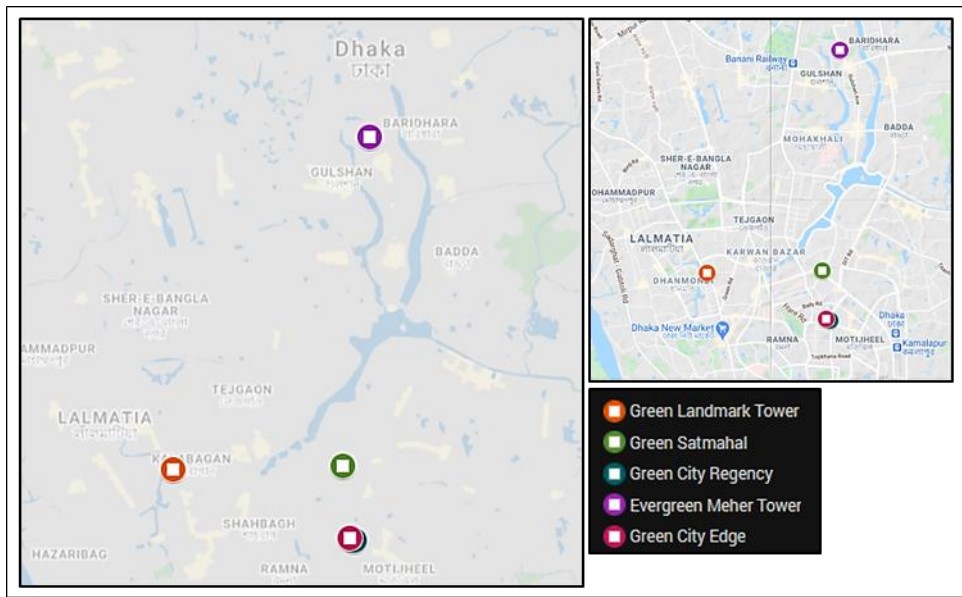

**Figure 1.** Map and location of selected commercial buildings in Dhaka.

*2.2. Data Sources*

Historical daily rainfall data for 28 years (1988–2015) for Dhaka were collected from the Bangladesh Meteorological Department (BMD). The rainfall data were analyzed for annual rainfall for each year and also for the average annual rainfall over the 28-year period. The model was run for three different climate scenarios, namely wet year, dry year, and normal year. The years having the highest (2885 mm) and the lowest (1169 mm) annual rainfall were considered as the wet year (2007) and the dry year (1992), respectively. The year with an annual rainfall close to the 28 years' average annual rainfall (2044 mm) was considered as the normal year (1996).

Figure 2 illustrates the annual rainfall (mm) in each year. Taking into consideration a 150 mm deviation from the average annual rainfall, the normal years occurred 7 times (1990, 1996, 1997, 2000, 2002, 2006, 2008). Accordingly, having higher values from the average was considered as the wet years, which occurred 9 times (1988, 1991, 1993, 1998,

1999, 2004, 2005, 2007, 2015). The dry years occurred 12 times (1989, 1992, 1994, 1995, 2001, 2003, 2009, 2010, 2011, 2012, 2013, 2014) in this 28-year period.

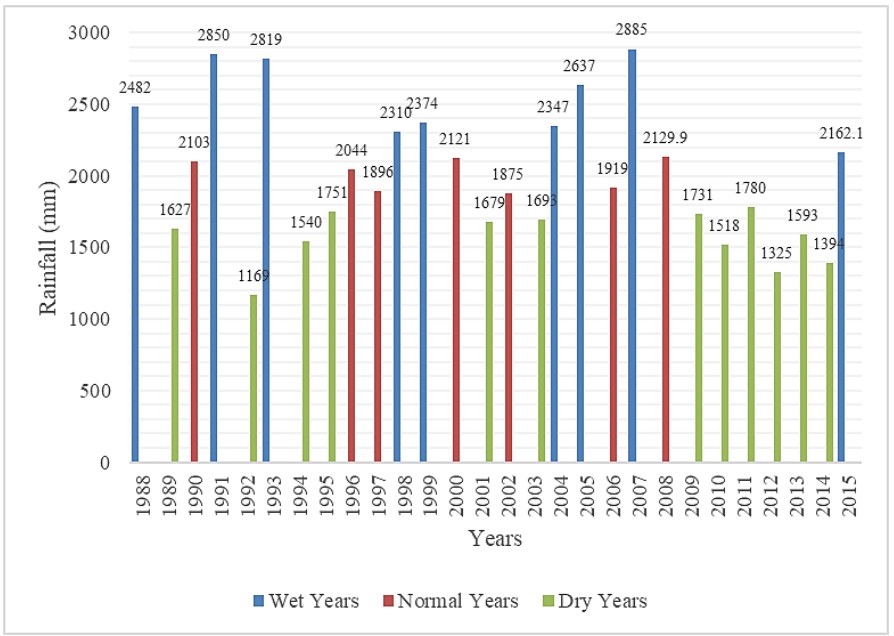

**Figure 2.** Annual rainfall variations (1988–2015) in Dhaka.

For the current study, a field survey was conducted to collect information regarding the roof catchment areas and underground reservoir capacity of five commercial buildings located in different parts of Dhaka. The roof catchment areas of the five buildings were as follows: 315 m$^2$ (Evergreen Meher Tower), 452 m$^2$ (Green Landmark Tower), 532 m$^2$ (Green Satmahal), 562 m$^2$ (Green City Regency), and 727 m$^2$ (Green City Edge). The capacities of the underground reservoirs to store the city supply water of these buildings were 162 m$^3$, 109 m$^3$, 114 m$^3$, 324 m$^3$, and 566 m$^3$, respectively.

The reservoir in each building was also considered as a rainwater tank to store the runoff volume of rainfall on the roof catchment (Figure 3). In general, the catchment area of commercial buildings in Dhaka normally ranges from 300 to 800 m$^2$ and the storage reservoirs range from 100 to 600 m$^3$. According to the BNBC [24], the net usable area of 15 m$^2$ per person is required for a commercial building and, accordingly, the total number of occupants of each of the commercial building was estimated. According to the BNBC [24], non-potable daily water demand ranges from 30 to 45 L per capita per day (lpcd), which was considered in this analysis. The runoff volume from the rooftop catchment area was calculated on a daily basis and runoff coefficients ranging from 0.8 to 0.9 were considered to account for various losses from net precipitation. Table 1 provides the survey data of occupants of the commercial buildings, catchment size, as well as installation and maintenance costs that were used in the model.

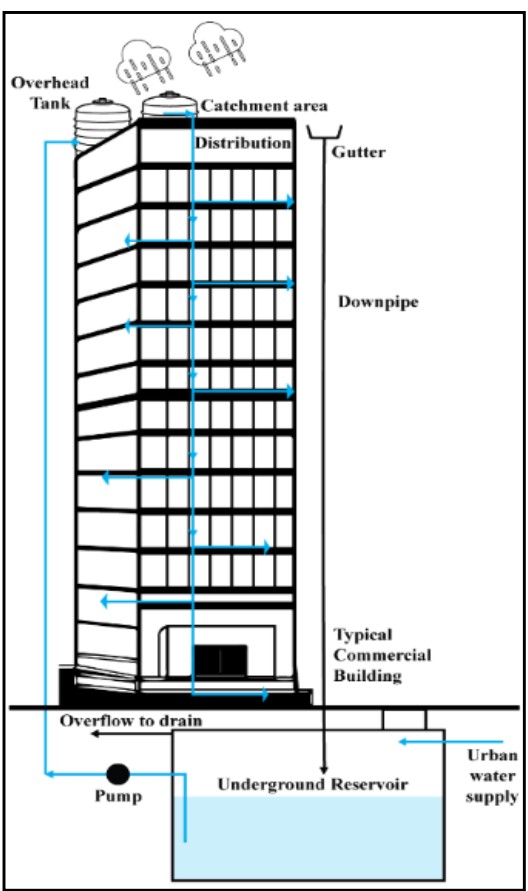

**Figure 3.** Typical commercial building and model assumption in a non-potable water supply system.

**Table 1.** Surveyed data of the commercial buildings (catchment area, commercial activities, floors, occupants, tank size, installation and maintenance costs).

| Building Names | Catchment Area (m²) | Commercial Activities | Total Floors | Total Occupants | Tank Size (m³) | Installation Costs (BDT) * | Maintenance Costs (BDT/Year) ** |
|---|---|---|---|---|---|---|---|
| Evergreen Meher Tower | 315 | Offices, restaurant | 2 Basement +14 | 294 | 162 | 30,000 | 10,000 |
| Green Landmark Tower | 452 | Offices, doctors' offices | 2 Basement +13 | 392 | 109 | 40,000 | 13,000 |
| Green Satmahal | 532 | Offices, bank, and restaurant | 2 Basement +14 | 496 | 114 | 45,000 | 15,000 |
| Green City Regency | 562 | Offices, bank, and food court | 2 Basement +22 | 824 | 324 | 50,000 | 18,000 |
| Green City Edge | 727 | Offices, bank, and restaurant | 2 Basement +15 | 727 | 566 | 60,000 | 20,000 |

BDT, Bangladeshi taka. * Installation costs include the cost of downpipe (PVC) and installation costs of downpipe and other fittings to convey the runoff into the underground reservoir, not the cost of construction of the underground reservoir. ** Maintenance costs include annual maintenance of downpipe and other fittings and cleaning of the tank.

Water demand in the commercial sector is mainly non-potable (used for cleaning of floors and parking lots, toilet flushing, etc.), which does not require water to meet the water quality as potable water. Rainwater is considered as the purest water and, if

harvested properly, it does not contain any health hazard substance; this water can supply for non-potable purposes without posing any significant health risk. The portion of the supply water used for drinking purposes is generally treated by water purifiers by each office section in the building; treatment of the harvested rainwater is not generally required and was not considered in calculating the annual maintenance costs.

Operational and maintenance costs were considered for a regular cleaning purpose once in a year. As the underground water reservoir also holds the rainwater, the maintenance costs were split between the rainwater and DWASA-supplied water. A total water consumption of 30–40 lpcd (which includes potable water demand) based on the BNBC [24] for the commercial buildings was considered in the analysis. For economic analyses, the unit water price was considered as 37.04 Bangladeshi taka (BDT) per kl of water (as per DWASA 2019 commercial water price). For energy consumption and savings, the procedure outlines by Anwar [30] were followed. According to Anwar [30], DWASA required electric energy of 0.3 kWh to produce 1.0 m$^3$ of water and the electricity price was assumed to be 4.12 BDT/kWh according to DWASA [1]. It was also assumed that the building owners would get the rebate for energy saving equivalent to the water saving obtained by rainwater harvesting.

### 2.3. Water Balance Model

A daily water balance model based on daily water input and daily non-potable water demand was developed in MATLAB R2018a version 9.4 by MathWorks to analyze the performance of the rainwater harvesting system, considering the factors associated with an RWH system (rainwater tank volume, daily rainfall and losses, daily water demand, and daily overflow loss). The water balance model was developed based on a behavioral type model as described by Imteaz et al. [31] and Karim et al. [16]. For better visualization and comparison, wide ranges of catchment areas (315 m$^2$, 452 m$^2$, 532 m$^2$, 562 m$^2$, 727 m$^2$) and storage tank capacities (100 to 600 m$^3$) were considered for the model simulations. Details of catchment areas and tank sizes are provided in Table 1. The model was run for three distinctive climatic scenarios (wet, dry, and normal year) as discussed earlier. The model calculates the annual water and energy savings, monetary savings, and payback period with a benefit–cost (B/C) ratio of a specific tank volume, taking into account the water price, operation and maintenance costs, project life cycle, and internal rate of return. In this study, the existing underground reservoir constructed for the storage of city water supply in each building was used as rainwater storage tank, so the construction of an extra tank was not required. In the model, rainwater was diverted to the underground reservoir during the rainfall and was used to fulfill the non-potable water demand of the commercial buildings first. If the rainwater volume was inadequate to meet the demand, then the city water supply was used to meet the excess demand. In the model, both time-based reliability and volumetric reliability were calculated as defined below. In addition, overflow/spillage was calculated when the accumulated runoff volume exceeded the tank (i.e., storage) capacity. Furthermore, an economic analysis section was included to convert water and energy savings into monetary benefits, as this would be the final decision-making outcomes to be used to convince the end-users such as building owners.

Like other similar studies [16,32–35], the model calculates the amount of water stored, water used, urban water supply required,. and overflow at the end of each time step (daily basis). All these data are collected to generate the outcomes on an annual scale. The mathematical equations of the water balance simulation model are outlined below:

$$A_t = S_t + A_{t-1} - D_w \tag{1}$$

$$A_t = 0, for\ A_t < 0 \tag{2}$$

$$A_t = T, for\ A_t > T \tag{3}$$

where $A_t$ is the accumulated rainwater (l) stored in the tank at the end of the tth day, $S_t$ is the stored rainwater volume (l) on the tth day, $A_{t-1}$ is the remaining rainwater volume (l)

in the tank at the beginning of the tth day, $D_w$ is the daily water demand (l) on the tth day, and $T$ is the underground reservoir capacity (l). In each time step, the equations used to calculate the volume of spilled water ($W_S$), town water supply ($W_T$), and rainwater used ($W_R$) are given by the following:

$$W_S = S_t + A_{t-1} - D_w - T, \, for \, S_t + A_{t-1} - D_w > T \tag{4}$$

$$W_T = D_w - S_t - A_{t-1}, \, for \, S_t + A_{t-1} < D_w \tag{5}$$

$$W_R = D_w, \, for \, S_t + A_{t-1} > D_w \tag{6}$$

$$W_R = S_t + A_{t-1}, \, for \, S_t + A_{t-1} < D_w \tag{7}$$

To achieve a more optimistic calculation, "yield after storage (YAS)", i.e., water yield, was calculated after the calculation of spillage and available storage. Lade et al. [36] reported that the YAS algorithm underestimates available water, whereas the "yield before storage (YBS)" algorithm has a tendency to overestimate the available supply.

In the model, the overflow ratio is defined as the ratio of the spilled rainwater from the tank (in the case of rainwater volume exceeding the tank storage capacity) to the rainwater inflow (diverted from the roof catchment) to the tank, as follows:

$$\text{Overflow ratio} = \frac{\sum W_S}{\sum S_t} \times 100 \tag{8}$$

where $\sum W_S$ is the total volume of rainwater spilled and $\sum S_t$ is the total volume of rainwater diverted during the calculation period.

Both time-based reliability ($R_T$) and volumetric reliability or efficiency of the system ($R_V$) were calculated in the model. The time-based reliability is defined as follows:

$$RT = \frac{T_D - U_D}{T_D} \times 100 \tag{9}$$

where $T_D$ is the total number of days in a year and $U_D$ is the total number of days in a year when collected rainwater fails to meet the intended demand. The volumetric reliability was calculated as follows:

$$RV = \frac{Total \, annual \, rainwater \, supply \, volume}{Total \, annual \, water \, demand} \times 100 \tag{10}$$

*2.4. Economic Analysis*

For the economic analysis, the annual water savings were calculated and then the annual energy saving, $E_S$, was calculated as follows:

$$E_S = AWS \times \alpha \tag{11}$$

where *AWS* is the annual water savings and $\alpha$ is the average energy required per unit of water production.

The annual monetary savings were calculated as follows:
Annual monetary savings for water,

$$MAWS = AWS \times P_W \tag{12}$$

Annual monetary savings for energy,

$$ME_S = E_S \times P_E \tag{13}$$

Total annual monetary

$$savings = MAWS + ME_S \tag{14}$$

where $P_W$ is the unit price of water and $P_E$ is the unit energy price.

Payback period (in years) for the RWH was calculated by comparing the cumulative annual monetary savings ($MAWS + ME_S$) in successive years with the total investment (initial installation cost + cumulative operation and maintenance costs in successive years) of the RWH system. Net savings for a year are the annual monetary savings subtracted by the annual operation and maintenance costs. For the calculation of net savings, an average savings value from the three weather conditions (wet, normal, and dry) was considered, assuming that a mix of dry, normal, and wet weather conditions would prevail. Net savings under different scenarios for the successive years were converted to equivalent net present value (NPV) as per the following equation:

$$\text{NPV of Savings} = \frac{Annual\ Savings}{(1+r)^n} \tag{15}$$

where *Annual Savings* is the savings for a year, *r* is the internal rate of return and *n* is the number of years since installation. As the calculated return periods were quite low, payback periods were calculated for an increment of 0.25 year.

Economic benefit is considered as the key determinant that convinces the building owners to construct an RWH system in the building. It is determined by the benefit–cost ratio, which is the ratio of the net present value of benefits (*B*) to the net present value of costs (*C*) associated with the RWH system. In the case where the benefit–cost ratio is more than 1.0, the RWH system is economically beneficial as the benefits provided by the system are more than the costs. The system will be more feasible, if the benefit–cost ratio is higher. The benefit–cost ratio was calculated as follows:

$$\text{Benefit–cost ratio} = \frac{B}{C} \tag{16}$$

where *B* represents the net present value (NPV) of the benefits and *C* represents the NPV of the costs. Equations related to benefits and costs for the economic analysis of water and energy savings are given below:

$$B = X \times \frac{(1+r)^{1+n} - 1}{r(1+r)^n} \tag{17}$$

$$C = IC + OM \times \frac{(1+r)^{1+n} - 1}{r(1+r)^n} \tag{18}$$

where *r* is the internal rate of return (%/year), *n* is the project life in years, *IC* is the investment cost for the RWH system, *OM* is the annual operational and maintenance cost, and $X = MAWS + ME_S$ or annual monetary benefits from water and energy savings. All costs and benefits were calculated based on Bangladesh currency, BDT (Bangladeshi taka).

*2.5. Uncertainities, Shortcomings, and Scope of the Study*

The main uncertainty in the mentioned methods is that for the economic analysis, annual water savings were taken as the averages of the three year types' (wet, normal, and dry) water savings, i.e., it considers three different weather scenarios occurring consecutively. In reality, it may not be the case, as weather does not follow such alternate trend. However, for a longer period, such an assumption will be close to reality. Additionally, the considered value of "internal rate of return" may change over the course of time, although under the current economic conditions it does not change drastically.

For the current study, a physically based mathematical model was used applying climate extreme values (i.e., rainfalls). Because climate extremes outcomes are expected to widely vary, however, these variations do not warrant any statistical analysis as model outcomes are all deterministic.

## 3. Results and Discussion

### 3.1. Reliability Analysis

The time-based reliability as calculated using Equation (9) for various tank volumes for each building under three climatic conditions for a non-potable water demand of 30 lpcd is shown in Figure 4. The time-based reliability depends on catchment area, rainwater demand, and climate condition. The building with a catchment area of 452 m$^2$ shows the highest reliability under both wet and normal climate scenarios, whereas the building with a catchment area of 562 m$^2$ shows the lowest reliability. Higher reliability can be attained for all buildings under a wet year. Under a dry year, the buildings with catchment areas of 452 m$^2$ and 532 m$^2$ have the highest reliabilities. For all the buildings (i.e., irrespective of catchment area), the time-based reliabilities do not change with the change in tank volume (especially for tank volumes greater than 100 m$^3$) under all the three climate scenarios. The main reason for this is due to the dominant seasonal pattern of rainfalls (most of the rainfalls occur during monsoon, whereas almost no rainfall occurs during winter), which significantly affects the time-based reliability. Figure 4a shows that the reliability under the wet climate varies from 8% to 21%, and Figure 4b shows that the reliability under the normal climate varies from about 4% to 10%. The reliability of the RWH system for the commercial buildings under dry weather is very low, only about 1% to 4% as shown in Figure 4c.

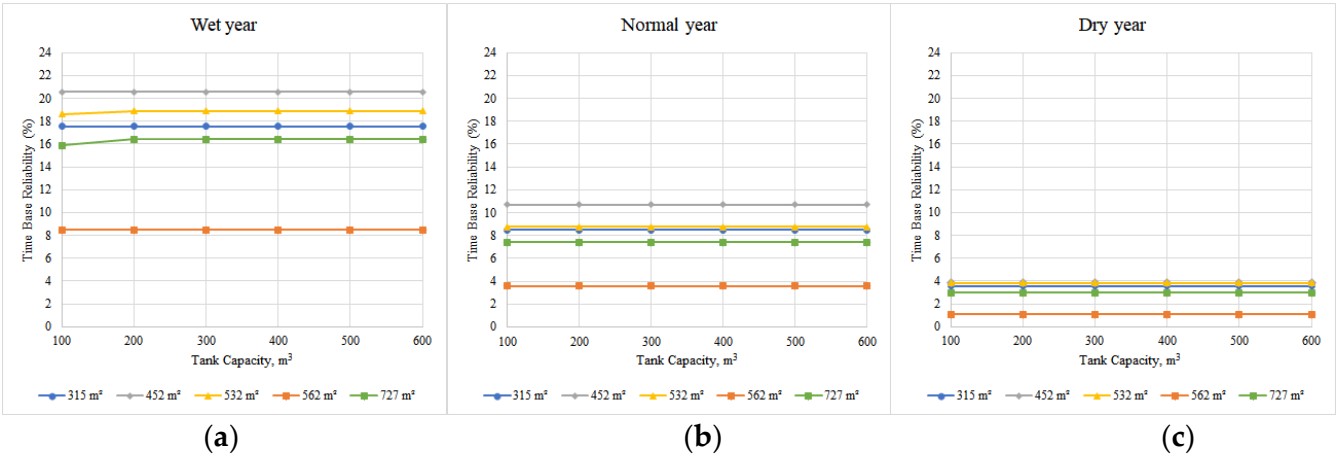

(**a**) (**b**) (**c**)

**Figure 4.** Time-based reliability curves with varying tank sizes for (**a**) wet, (**b**) normal, and (**c**) dry year climate scenarios.

Figure 5 shows the volumetric reliability of the RWH system with tank volume for each building. Volumetric reliability, which also indicates the percentage of water savings, follows the similar pattern of the time-based reliability curves as shown in Figure 4, but with higher reliability for the same building and it does not change beyond the tank size of 100 m$^3$. It was observed that the time-based reliability of the RWH in the buildings varies from 8% to 21% under the wet climate scenario as shown in Figure 4a, but the volumetric reliability differs from 16% to 27% for the similar climatic condition as shown in Figure 5a. Under the normal climate scenario, about 11% to 19% of the non-potable water demand of the buildings can be supplemented from the RWH system as shown in Figure 5b. Highest volumetric reliability can be obtained for the wet climate condition and a maximum of 27% water can be supplemented under this condition. Under the dry climate scenario, more than 6% water can be supplemented from the RWH system of the commercial buildings as shown in Figure 5c.

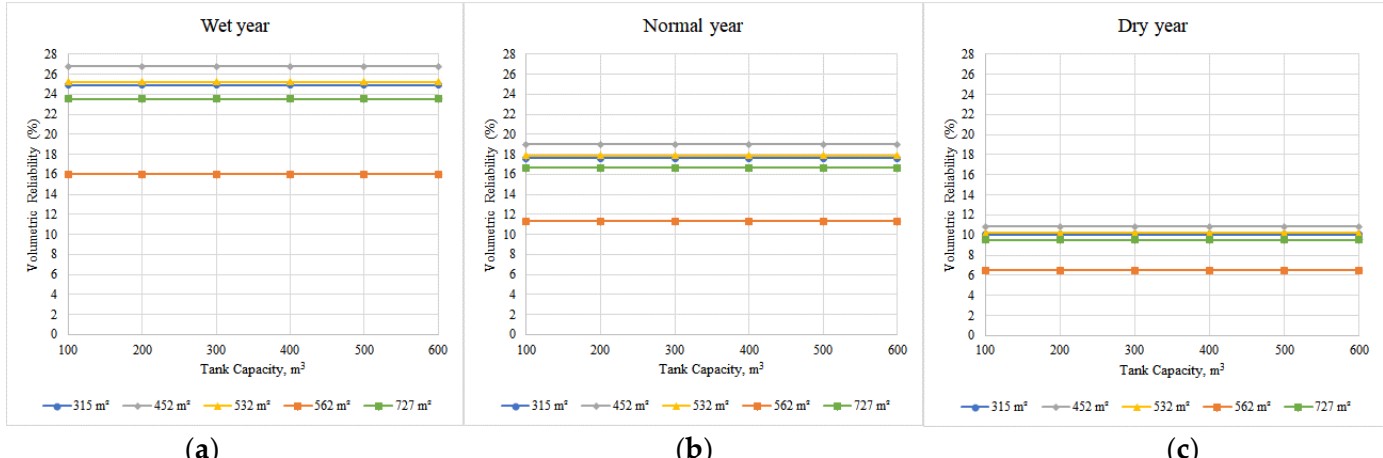

**Figure 5.** Volumetric reliability curves with varying tank sizes for (**a**) wet, (**b**) normal, and (**c**) dry year climate scenarios.

Figure 6 shows the time-based reliability curves against per capita water demand of the RWH system of the buildings with their corresponding storage tank capacity as mentioned in Table 1. These time-based reliability curves again show that the building with a catchment area of 452 m$^2$ has the highest reliability under both wet and normal climate scenarios, whereas the building with a catchment area of 562 m$^2$ has the lowest reliability. The time-based reliability decreased by about 30% to 60% under the wet climate condition, when the demand increased from 30 lpcd to 45 lpcd, as shown in Figure 6a. Figure 6b,c shows that, under both dry and normal climate conditions, this reliability decreased by 25% to 50%. Volumetric reliability for various water demands of the buildings are shown in Figure 7. These curves follow the same trend as time-based reliability and decrease with increase in water demand. For these commercial buildings, the volumetric reliability under the wet climate condition varies from 10% to 18% for a water demand of 45 lpcd and from 16% to 27% for a water demand of 30 lpcd as shown in Figure 7a. With an increase in water demand from 30 to 45 lpcd, the volumetric reliability was found to decrease by more than 60% under all climate conditions.

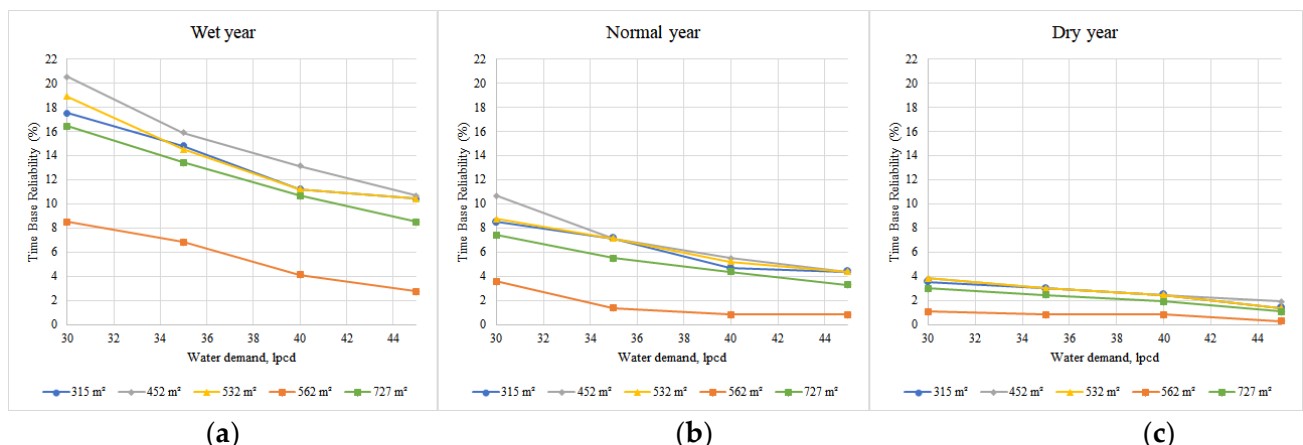

**Figure 6.** Time-based reliability curves with varying water demands for (**a**) wet, (**b**) normal, and (**c**) dry year climate scenarios.

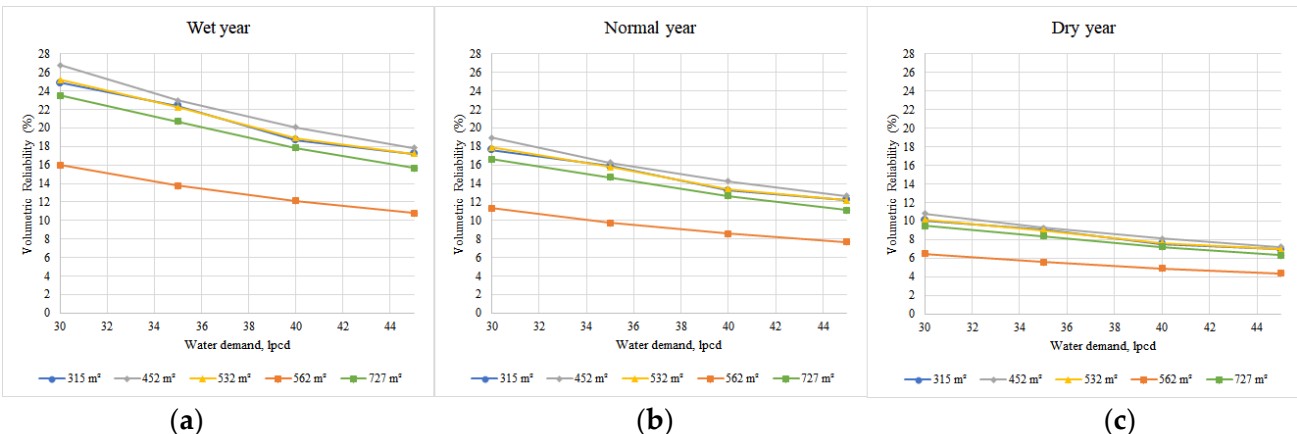

**Figure 7.** Volumetric reliability curves with varying water demands for (**a**) wet, (**b**) normal, and (**c**) dry year climate scenarios.

*3.2. Overflow Ratio*

The overflow ratio indicates the percentage of the spilled rainwater to the rainwater collected from the roof catchment to the system. The overflow ratio of the RWH system of the buildings with a water demand of 30 lpcd under wet climate conditions is shown in Figure 8.

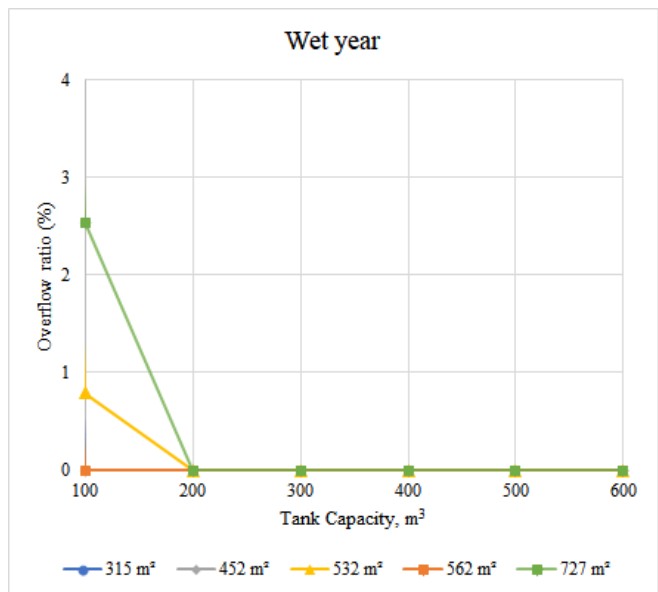

**Figure 8.** Overflow ratio with varying tank sizes under the wet year climate condition.

It can be seen that very insignificant overflow would occur from the RWH systems when the tank size is more than 100 m$^3$ under the wet climate condition and no overflow would occur for the larger tanks. Since the minimum capacity of the storage tank of the buildings is 109 m$^3$ (Table 1), very insignificant overflow would occur from the storage tanks under the wet climate condition and no overflow would occur from the storage tanks under both dry and normal climate conditions. Thus, the existing reservoir would be adequate to control overflow of the harvested rainwater and, in the case of a larger catchment with a storage tank <100 m$^3$, overflow might occur under the wet climate condition. This analysis reveals that integrating an RWH system in commercial buildings would act as a detention tank, which would eventually alleviate the water clogging problem frequently caused by heavy monsoonal rainfalls. A previous analysis by Karim et al. [16]

also confirmed that no significant overflow would occur during the monsoon from the existing water reservoirs of the residential buildings in Dhaka, if RWH was practiced in the residential buildings.

### 3.3. Water Use Pattern

In Dhaka, a huge rainfall occurs every year during the monsoon period between June and October. Figure 9 shows the quantity of the monthly water demand that can be augmented by RWH under the normal year climate condition for a building with a catchment area of 452 m$^2$ and a storage tank capacity of 109 m$^3$ for a non-potable water demand of 30 lpcd.

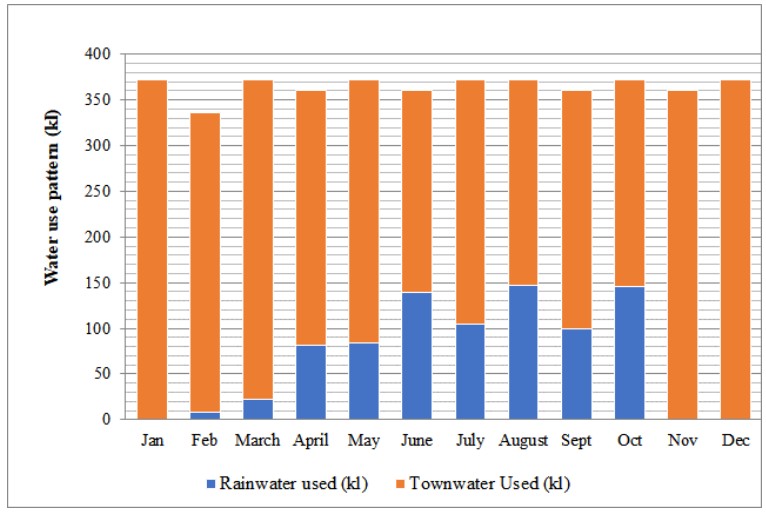

**Figure 9.** Monthly water demand supplemented by rainwater harvesting (RWH) under the normal year climate condition for a building with a catchment area of 452 m$^2$.

As shown in Figure 9, the percentages of monthly water demand that can be supplemented by rainwater harvesting during May to October were about 22% to 40%. Under the wet year climate condition, this supplemented amount was about 20% to 82%. Similar results were observed for other buildings. The analysis results reveal that, for a water demand of 30 lpcd, about 11% to 19% of the yearly water demand can be met by RWH under the normal year climate condition and that, under the wet year climate condition, harvested rainwater can meet about 16% to 26.8% of the annual water demand of the buildings (Table 2), which leads a significant water saving of the urban piped water supply.

### 3.4. Economic Analysis

Economic analysis is important for decision-making and for evaluating the implementation of a rainwater harvesting system. The economic viability of the RWH in the commercial buildings was calculated for annual monetary savings for water and energy, payback period, and benefit–cost ratio under three climate scenarios.

Considering a water demand of 30 lpcd and 0.3 kWh/m$^3$ of energy consumption for water production, annual savings of both the water and energy through RWH in the commercial buildings are shown in Table 2. It reveals that both the monetary and energy savings increase with the increase in catchment area under the three climate scenarios; about 580 kl of water and 174 kWh of energy can be saved for a catchment area of 315 m$^2$ under the average climate condition. These values increased to about 1338 kl of water and 401 kWh of energy for a catchment of 727 m$^2$. As expected, a relatively small amount of water can be supplemented under the dry climate condition. Table 2 also shows the percentage of water saved/supplemented through the rainwater tank, which is basically the ratio of annual water savings to total annual water demand for each building.

**Table 2.** Annual water and energy savings under wet, normal, and dry year climate conditions.

| Catchment Area (m²) | Climatic Condition | Annual Savings | | % of Water Supplemented from Rainwater |
| --- | --- | --- | --- | --- |
| | | Water (kl/Year) | Energy (kWh/Year) | |
| 315 | Wet year | 817.90 | 245.37 | 24.90 |
| | Normal year | 579.47 | 173.84 | 17.64 |
| | Dry year | 331.41 | 99.42 | 10.09 |
| 452 | Wet year | 1173.62 | 352.09 | 26.80 |
| | Normal year | 831.50 | 249.45 | 18.98 |
| | Dry year | 475.55 | 142.67 | 10.86 |
| 532 | Wet year | 1381.34 | 414.40 | 25.23 |
| | Normal year | 978.67 | 293.60 | 17.88 |
| | Dry year | 559.72 | 167.92 | 10.22 |
| 562 | Wet year | 1459.23 | 437.77 | 16.00 |
| | Normal year | 1033.86 | 310.16 | 11.33 |
| | Dry year | 591.28 | 177.38 | 6.48 |
| 727 | Wet year | 1887.66 | 566.30 | 23.50 |
| | Normal year | 1337.39 | 401.22 | 16.65 |
| | Dry year | 764.88 | 229.46 | 9.53 |

### 3.5. Payback Period

Using the annual (water plus energy) monetary savings and the overall expenditures (installation plus operation and maintenance costs) and converting future net savings to the "net present value", payback periods were calculated for rainwater harvesting in each building for the three climatic conditions. For the calculation of net present values, based on an earlier study by Karim et al. [16] an internal rate of return of 12.5% was considered. The outcomes of the payback period analysis are presented in Table 3.

**Table 3.** Monetary savings, payback periods, and benefit–cost (B/C) ratios for different catchment areas.

| Catchment Area (m²) | Annual Monetary Savings Water + Energy, (BDT/Year) | Payback Period Water + Energy, (Year) | Benefit–Cost Ratio Water + Energy |
| --- | --- | --- | --- |
| 315 | 22,057 | 3.75 | 1.5 |
| 452 | 31,650 | 3.25 | 1.7 |
| 532 | 37,252 | 3.00 | 1.7 |
| 562 | 39,352 | 2.75 | 1.5 |
| 727 | 50,906 | 2.50 | 1.8 |

The calculated payback periods are found to be quite short, varying from 2.25 to 3.75 years, depending on the catchment area of the buildings and the climatic condition. This payback period was calculated based on the current water price of DWASA. These payback periods are likely to be even shorter under the situation of a higher water price in the future for commercial uses. As compared with rainwater harvesting for the residential buildings in Dhaka, the payback periods were 4.6 years and 7.3 years under the wet and normal weather conditions, respectively [9], revealing the fact that rainwater harvesting for commercial buildings is more beneficial and has return periods that are almost half the return periods for residential buildings.

### 3.6. Benefit–Cost Ratio

Considering the economic savings for both water and energy as benefits and installation, operation, and maintenance costs as costs, the benefit–cost ratio of the RWH in the commercial buildings for a project life of 10 years under the three climate conditions was

calculated and is shown in Table 3. In this analysis, a 12.5% internal rate of return was assumed based on a previous study by Karim et al. [16]. To calculate the benefit–cost ratio, an average savings from all three weather conditions were considered, as it is unlikely that a certain weather condition (i.e., dry) will persist for a long time; rather it is reasonable to expect that a mix of weather conditions (dry, normal, and wet) will occur. As expected, the benefit–cost ratio generally increases with an increase in catchment area, and a maximum benefit–cost ratio of 1.8 was observed for the building with a catchment area of 727 m$^2$. For this large roof, a benefit–cost ratio of more than 1.0 is achieved, even if all future years happen to be dry, which is very rare in a tropical country like Bangladesh. A 10-year project life of an RWH system in a commercial building is too short as compared to the building life, which is about 50 years or more. The benefit–cost ratio increases by considering the project life of the RWH system as more than 10 years. In recent years, it has been observed that water price including commercial water price was increased in regular intervals by DWASA, which means that the monetary savings will be more under future water prices and that the introduction of RWH in a commercial building will be economically more viable and feasible.

The projected population of Dhaka is expected to reach 32 million in 2035 as compared to the present population of about 20 million. To supply adequate water to this vast population, DWASA would need a huge investment in sourcing, treating, and supplying the required water to the city dwellers. As discussed, rainwater harvesting both in commercial and residential buildings is one of the favorable alternative water supplies, especially for non-potable purposes, as it seemed feasible both economically and environmentally for a city like Dhaka. The installation of RWH systems in commercial buildings thus contributes several environmental benefits by reducing dependency on urban pipe water supply, which mainly comes from groundwater. It contributes to the substantial reduction of the urban water logging problem and adds significant monetary savings for the building owners. The findings in this study would be useful for local government authorities, policy makers, and house owners to understand the importance of RWH in commercial buildings in terms of environmental and economic benefits.

## 4. Conclusions

In this paper, a methodology for analyzing the reliability and economic benefit of rainwater harvesting in commercial buildings in an urban context was presented and illustrated through a case study for the capital city of Bangladesh, i.e., Dhaka. The analysis was conducted for three climate scenarios (wet, dry, and normal year) with data collected for five commercial buildings having catchment areas varying from 315 to 776 m$^2$ and a storage tank capacity varying from 100 to 600 m$^3$. In the past, several studies focused on residential RWH; however, this is the first study on RWH for the commercial buildings in the country (i.e., Bangladesh). Corporate authorities pay minimum attention to the implementation of RWH in commercial buildings, whereas the local water supply authority, DWASA, is trying to find alternative sources of water to supply due to a tremendous increase in water demand. This paper provides an insight into the potentials of monetary and energy savings by implementing rainwater harvesting in commercial buildings and will increase awareness among stakeholders, who will eventually take necessary steps towards implementing such a sustainable feature.

The results of the analysis reveal that both time-based reliability and volumetric reliability do not change beyond the tank capacity of more than 100 m$^3$. Under the normal year climate scenario, the potential savings of water from 500 to 1337 kl and energy from 174 to 401 kWh per year can be achieved and about 11% to 19% water can be supplemented from the RWH system. Under the wet climate scenario, both water and energy savings are increased significantly and a maximum of 27% of water demand can be supplied from the RWH system. Under the dry climate condition, more than 6% water can be supplemented from the RWH system to satisfy the non-potable water demand of the buildings. Very insignificant overflow was found to occur from the storage reservoir under the wet year

climate scenario, and no overflow would occur under both dry and normal year climate conditions. Thus, RWH in commercial buildings can have a positive impact on urban water stress through water savings and can also counteract overflow of the monsoon rainfall responsible for the severe water clogging problem during heavy rainfalls in Dhaka.

The economic analysis of the RWH system indicates a short payback period from only 2.25 to 3.75 years, depending on the catchment area and climatic condition. Considering a mix of dry, normal, and wet years, RWH in commercial buildings is economically beneficial and feasible, even for a roof size of 315 m$^2$ (the smallest size considered in this study). Calculated payback periods turned out to be short due to the fact that water and energy prices are higher for the commercial sector compared to the residential sector. It is to be noted that, with the economic growth of the country, those prices are expected to keep on increasing in the future, and such an RWH system is expected to provide more benefits to the users and the authorities. This study will encourage policy makers to enact the necessary legislative measures for the implementation of an RWH system in commercial buildings, which will bring considerable economic and environmental benefits, as well as reduce stress on the urban pipe water supply. For the payback period calculations. consecutive dry, normal, and wet years were considered; however, in reality, the weather may not follow such a regular pattern.

It is possible to generalize the expected payback period associated with underlying factors such as roof area, tank size, and interest rate, which will be a part of future study. Furthermore, as the findings of such a study depend on several climatic factors such as rainfall amount and seasonal pattern, the presented outcomes may not be valid for other regions or cities. In general, an individual city or region needs to conduct an individual study depending on specific rainfall amounts and patterns, unless a city or region is very similar to other city or region with regard to the rainfall amount and pattern. Nonetheless, one may endeavor to find generalized outcomes incorporating rainfall amounts and patterns, in addition to the basic factors such as roof area and tank size.

**Author Contributions:** Conceptualization, M.R.K. and M.A.I.; methodology, B.M.S.S. and S.S.S.; validation, M.R.K., B.M.S.S. and S.S.S. formal analysis, S.S.S.; investigation, B.M.S.S.; resources, B.M.S.S.; data curation, B.M.S.S.; writing—original draft preparation, B.M.S.S.; writing—review and editing, M.R.K. and M.A.I.; visualization, B.M.S.S.; supervision, M.R.K. and M.A.I.; project administration, M.R.K. and M.A.I.; funding acquisition, M.A.I. All authors have read and agreed to the published version of the manuscript.

**Funding:** This research received no external funding.

**Institutional Review Board Statement:** Not applicable.

**Informed Consent Statement:** Not applicable.

**Data Availability Statement:** The data presented in this study are available on request from the corresponding author. The data are not publicly available due to the restriction from the data owner.

**Conflicts of Interest:** The authors declare no conflict of interest.

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
