# Peer review of "Rainwater Harvesting Potentials in Commercial Buildings in Dhaka: Reliability and Economic Analysis"

_hydrology, doi:10.3390/hydrology8010009_

Round 1

Reviewer 1 Report

Generally, I recommend this manuscript for potential publication after minor revision. My concerns are listed as follows,

  • The abstract is too simple. The scientific importance, new findings, and limitation of this study should be highlighted.
  • liter per capita per day, abbreviated as lpcd, should be given the full form when it firstly appears in the abstract.
  • In section 3.3. Economic analysis, how about the annual expense of maintaining the rainwater harvesting facilities? For example, repairing and replacing the aging pipes?
  • For Figures 2-6,are there significant differences in reliability curves with varying tank sizes?I mean at least the ANOVA method should be used to validate the results.

Author Response

Provided in the attached file. 

Reviewer 2 Report

The paper presents an interesting analysis of the reliability and economic benefit of rainwater harvesting in commercial buildings, taking the city of Bangladesh as case study. It is a suitable topic for Hydrology MDPI journal. The manuscript is professionally written, clear, and easy to read. The results are relevant, well presented and discussed.

I recommend a revision of the manuscript following my comments below.

  • Line 40: It is the first time that “RWH” appears in the text, and it has not been explained or fully cited before.
  • Line 78: It is the first time that “BNBC” appears in the text, and it has not been explained or fully cited before.
  • Table 1: Although implicit in the text before the table, the building names should be included in the Table.
  • Line 132: The MATLAB code could be provided in the manuscript.
  • Line 148: Which research?
  • Line 181: “annual”.
  • Why only the data of Moinho da Gamitinha was chosen? And why this one so far (approximately 45 km) from the estuarine area? This could be better explained in the text.
  • Lines 123-125: Where are the mentioned dams? How far are they from the measurement point? The authors should better develop and justify this sentence.
  • Sub-section 1.1 could be moved into Section 2;
  • Section 4: Although the results are well presented, they are not exactly fully discussed (especially sub-section 4.1).
  • Table 2: This table could be better (more) discussed in the text. For example, the last column was not cited.
  • Conclusions: The possibilities of using the findings of this manuscript in other case studies (not only Bangladesh) should be (more) explored. What is the novelty of this paper? What does it bring to the literature?

Author Response

Provided in the attached file. 

Reviewer 3 Report

This paper is fairly well written. The intent is good. Some study results may add to the existing knowledge. The following comments, however, may further enhance the readability of this manuscript:

  1. “Dhaka” may be used in place of “the capital city of” in the paper title.
  2. Some keywords duplicate the same as in the paper title. Other selections should be re-chosen.
  3. Please use “m/yr” instead of “meters/year”.
  4. All acronym names, such as “BNBC”, “lpcd” should be defined when first appear.
  5. Please unify the use of “L” and “l” for liter unit.
  6. Diagram of the annual rainfall variations for the 28 years study period may be helpful. Discussion on how many wet, dry, and normal years over this 28 years period is suggested.
  7. Please use the proper multiplication symbol (x) in all the equations.
  8. In equations 17 and 18, is it “(1 +r)n-1?
  9. The consistency of the reliability curves is very strange. Why 563 m2 catchment area has lower reliability than 315 m2 catchment area in Figures 2 and 3? Similarly with Figures 4 and 5?
  10. The supplemental effect as shown in Figure 7 and Table 2 is not convincing at all. The cost to purchase town water is very high in normal years, not to mention dry years.
  11. As such, the “cost” factor computed in this study is very misleading and should be re-assessed.

Author Response

Provided in the attached file. 

Round 2

Reviewer 3 Report

A great job in revising the original manuscript. However, a minor spell check is recommended to upgrade the revised manuscript to publishable standard. Besides, "Dhaka city" and "Dhaka city" should be unified.

Author Response

Thanks for the positive comment and acceptance. The thorough spelling check was conducted.